# Characteristic Findings of Infants with Transient Elevation of Acylcarnitines in Neonatal Screening and Neonatal Weight Loss

**DOI:** 10.3390/ijns11020033

**Published:** 2025-04-29

**Authors:** Sakura Morishima, Yumi Shimada, Yoriko Watanabe, Kenji Ihara

**Affiliations:** 1Department of Pediatrics, Oita University School of Medicine, Yufu-City 879-5593, Japan; sakura0624@oita-u.ac.jp (S.M.); yuminasu@oita-u.ac.jp (Y.S.); 2Department of Pediatrics and Child Health, Kurume University School of Medicine, Kurume 830-0011, Japan; york@med.kurume-u.ac.jp; 3Research Institute of Medical Mass Spectrometry, Kurume University School of Medicine, Kurume 830-0011, Japan

**Keywords:** very-long-chain acyl-CoA dehydrogenase deficiency, false-positive results, postnatal starvation, mass screening, acylcarnitines, amino acids

## Abstract

The detection of elevated long-chain acylcarnitine levels, particularly C14:1 and the C14:1/C2 ratio, during neonatal screening may indicate very-long-chain acyl-CoA dehydrogenase deficiency (VLCADD), although similar findings can result from postnatal starvation. We investigated the relationship between false-positive results, postnatal weight loss, and subsequent growth. Additionally, we explored potential diagnostic markers of postnatal starvation. The following neonates from Oita Prefecture (April 2014–March 2024) were included in this study: patients identified as false-positive for VLCADD (*n* = 19), patients with VLCADD (*n* = 3), and children negative in mass screening who completed their 3-year-old health check-up (*n* = 30). The false-positive group exhibited significant weight loss at blood sampling for neonatal screening. An acylcarnitine analysis showed significant increases in various short- to long-chain fatty acids in the false-positive group, likely owing to enhanced fatty acid catabolism via β-oxidation. Elevation of a broad range of fatty acids and reduced amino acid levels seemed to be associated with significant weight loss at blood sampling.

## 1. Introduction

Breastfeeding has been globally recognized for its numerous advantages, including an optimal composition of nutrients that minimizes the metabolic burden on infants, a reduction in both the incidence and severity of infections, and a decreased risk of childhood obesity and the later development of type 2 diabetes. Additionally, breastfeeding offers benefits to mothers, such as facilitating postpartum recovery and fostering a strong mother–child bond [1]. According to the “Infant and Young Child Nutrition Survey” conducted by the Ministry of Health, Labour, and Welfare of Japan in 2015, over 90% of mothers expressed a desire to breastfeed, and more than 50% actually practiced breastfeeding. However, exclusive breastfeeding has been associated with nutritional concerns, including deficiencies in iron and vitamin D levels. In countries where early postpartum discharge is common, exclusive breastfeeding has been reported to increase the risk of hospital readmission due to remarkable weight loss and jaundice, raising concerns regarding both neonatal nutrition and childcare at home [2].

Very-long-chain acyl-CoA dehydrogenase deficiency (VLCADD) is a rare genetic disorder of fatty acid oxidation, characterized by the inability to properly break down long-chain fatty acids [3]. It typically manifests in early infancy, with acute episodes leading to cardiomyopathy and, in some cases, sudden unexpected death. Other patients experience chronic symptoms, such as muscle weakness, myalgia, and rhabdomyolysis [4]. Epidemiological studies based on newborn screening have reported that VLCADD is the most common fatty acid oxidation disorder, with an incidence of 0.07–1.9 per 100,000 individuals [4,5,6,7,8]. VLCADD can be identified with an acylcarnitine analysis by detecting elevated levels of long-chain acylcarnitines, specifically C14:1, and the C14:1/C2 ratio. Presymptomatic testing has been applied in neonatal mass screening programs in many countries [9,10,11,12]. However, transient neonatal starvation due to an insufficient supply of breast milk during the early postpartum period can result in mimicking the blood acylcarnitine pattern, as revealed by false-positive VLCADD [13].

This study aimed to investigate the effects of false-positive VLCADD results on subsequent infant growth and maternal commitment to neonatal nutrition, especially exclusive breastfeeding. We also explored the characteristic patterns of acylcarnitine and amino acid profiles in infants with transient neonatal starvation and false-positive VLCADD results. Finally, we presented potential markers of transient starvation during early infancy (Figure 1).

## 2. Materials and Methods

### 2.1. Neonatal Screening System in Oita Prefecture

Neonates whose parents provided informed consent underwent blood spot sampling on filter paper at 4–6 days of age at a maternity hospital in Oita Prefecture, Japan. The samples were directly mailed to a laboratory in Oita City (Almeida Laboratory Center) for analysis. Neonates requiring re-testing or further extensive examination were selected based on established cutoff values (Appendix A) under standard quality controls, and all results were reported to the maternity hospitals. If an acylcarnitine analysis detected any levels exceeding the cutoff value, the neonates were referred to a pediatric center for confirmatory testing (Department of Pediatrics, Almeida Hospital). Dried blood samples (DBSs) were collected and analyzed by tandem mass spectrometry without derivatization of the samples, as previously described [14]. Local pediatric experts of inborn errors of metabolism in Oita Prefecture discussed these cases and determined whether the case was positive or false-positive based on the results of confirmatory tests and additional biochemical analyses. Genetic testing was conducted to confirm the diagnosis for positive cases.

### 2.2. Study Subjects

From the stored tandem mass spectrometry data, 22 cases were beyond the cutoff values for C14:1 (>0.25) and the C14:1/C2 ratio (>0.013), suggesting VLDCAD. Further studies classified 19 patients as false-positive and 3 cases as positive. Among the 22 cases, 3 showed consistent results by multiple acylcarnitine analyses and underwent genetic testing, which confirmed a diagnosis of VLCAD. Two of them were siblings with a homozygous c.709T>C, p.C237R, a likely pathogenic variant, while one case exhibited compound heterozygous variants, c.1246G>A, p.Ala416Thr, a likely pathogenic variant, and c.864C>T, p.Phe288Phe, a likely benign variant. For the remaining 19 cases, subsequent acylcarnitine analyses and clinical courses indicated false positives of neonatal screening, and genetic testing was therefore not performed. We asked the parents if they were willing to participate in this study, and parental consent was obtained from 11 individuals in the false-positive group and no individuals in the positive group. For the remaining individuals—8 false-positive cases and 3 positive cases who did not respond to the survey request and from whom consent could not be obtained—only the data from the tandem mass spectrometry analysis were analyzed. This analysis was conducted after posting an opt-out notice on the institution’s website. For the healthy control group, 30 individuals were recruited from families of children who had completed vaccination prior to the 3.5-year health check-up and who gave their consent to participate in this study.

### 2.3. Ethical Considerations

This study was approved by the Ethics Committee of Oita University Faculty of Medicine (approval number: 2542).

### 2.4. Study Materials

#### 2.4.1. Extraction of Stored Data on Tandem Mass Spectrometry

Following approval from the ethics committee, data stored at the neonatal screening facility were reviewed to identify cases of neonates born in Oita Prefecture between April 2014 and March 2024 with C14:1 and C14:1/C2 levels exceeding the VLCADD screening cutoff values (C14:1 > 0.25, C14:1/C2 > 0.013), thus requiring further examination. Then, cases were classified as positive when the re-evaluation of dried blood spots was conducted by tandem mass spectrometry more than one month later, under sufficient milk feeding, and persistently elevated levels of C14:1 and C14:1/C2 were confirmed. In contrast, cases were classified as false-positive when all the results of two or more follow-up tests were within the normal ranges. Preterm infants, low-birth-weight infants, and those with underlying or comorbid conditions were excluded from the false-positive group. Requests were directly mailed to the parents of positive or false-positive cases to invite them to participate in this study. For those who provided their written consent, interviews were conducted either in person or by mail. If we could not contact the parents, an opt-out notice was posted on the research institution’s website; this study used minimal data extracted from medical records. Stored data on newborn screening of 30 healthy individuals were obtained as negative controls after the respondents provided their written informed consent for questionnaire-based research and the use of neonatal screening data.

#### 2.4.2. Selection of Healthy Controls for Interview

The parents of children who visited a pediatric hospital in Oita City (Oita Children’s Hospital, Oita City, Japan) for vaccination and who had completed their 3.5-year health check-up were invited to participate in this study. After obtaining written consent, we checked the maternal and child health records and interviewed the parents directly, mainly about breastfeeding.

### 2.5. Statistical Analysis

The data were analyzed using SPSS 24 (IBM, Tokyo, Japan). Normality was assessed using the Shapiro–Wilk test. Normally distributed data were compared using *t*-tests, whereas non-standardized distributed data were analyzed using the Mann–Whitney U test. Birth weight, weight at screening, and each value of items by tandem mass spectrometry were compared among the three groups using a one-way ANOVA with Tukey’s test for multiple comparisons (for normally distributed data) or the Kruskal–Wallis test followed by the Steel–Dwass test (for non-normally distributed data).

## 3. Results

### 3.1. Analysis of Study Subjects

The mean birth weights for the false-positive, positive, and negative groups were 3061.3 g, 3136.6 g, and 3052.0 g, respectively; all were near the mean values of Japanese children, with no significant differences among them. However, the mean body weights of these groups at the time of mass screening blood sampling were 2811.2 g, 3058.3 g, and 2946.0 g, respectively, with the false-positive group having the lowest weight. Correspondingly, the rate of weight loss was significantly greater in the false-positive group (−9.2%) than in the positive (−2.7%) and negative (−3.9%) groups. There were no significant differences in growth between the false-positive and negative groups at the 1-month health check-up (Table 1).

### 3.2. Comparison of Tandem Mass Spectrometry Analysis

#### 3.2.1. Comparison Between the Positive and Negative Groups

C14:1 (*p* = 5.1 × 10^−9^, two-group comparison) and C14:1/C2 (*p* = 5.1 × 10^−9^) levels were significantly elevated beyond the cutoff value in all the cases in the positive group. Additionally, regarding the diagnostic markers for glutaric acidemia type 2, such as C12 (*p* = 4.7 × 10^−5^), C14 (*p* = 1.5 × 10^−2^), or C14:2 (*p* = 2.0 × 10^−3^), the percentage of individuals whose values were outside of the pre-determined diagnostic cutoff values was significantly higher in the positive group. Conversely, C18:2 was significantly higher in the negative control group (*p* = 3.6 × 10^−2^). No significant differences were observed in other short- or medium-chain fatty acids. These findings confirm the relevance of these markers in the diagnosis of VLCAD (Table 2 and Table 3).

#### 3.2.2. Comparison Between the False-Positive and Negative Groups

Significantly higher levels of a broad range of acylcarnitines were observed in the false-positive group in comparison to the negative group. These included short-chain fatty acids, such as C4OH (*p* = 1.6 × 10^−7^) and C5-DC (*p* = 2.7 × 10^−5^), medium-chain fatty acids, such as C6 (*p* = 2.1 × 10^−5^), C8 (*p* = 4.6 × 10^−8^), C8:1 (*p* = 6.8 × 10^−6^), C10 (*p* = 5.0 × 10^−9^), C10:1 (*p* = 1.1 × 10^−7^), C12 (*p* = 5.1 × 10^−9^), and C12:1 (*p* = 1.9 × 10^−9^), and long-chain fatty acids, including C14 (*p* = 7.3 × 10^−8^), C14:2 (*p* = 1.9 × 10^−9^), C14OH (*p* = 1.0 × 10^−8^), C16 (*p* = 7.4 × 10^−6^), C16:1 (*p* = 2.1 × 10^−7^), C16-OH (*p* = 6.7 × 10^−8^), C18:2 (*p* = 2.3 × 10^−5^), C18OH (*p* = 5.6 × 10^−4^), and C18:1OH (*p* = 8.7 × 10^−8^). These findings suggest that fatty acid oxidation is accelerated by increased triglyceride degradation in a non-specific manner under nutritional deprivation (Table 2 and Table 3).

#### 3.2.3. Comparison Between the False-Positive and Positive Groups

A representative VLCAD marker, C14:1, was markedly elevated in both the false-positive and positive groups in comparison to the normal subjects, and no significant difference was observed between the two groups (Figure 2A). Additionally, the ratios of C14:1/C2 (*p* = 6.5 × 10^−4^) (Figure 2B) and C14:1/C16 (*p* = 1.7 × 10^−6^) (Figure 2C) were significantly higher in the positive group than those in the false-positive group. The overlap in these values between the groups suggests that these values were not sufficient to distinguish the positive results from the false-positive results.

Conversely, C6 (*p* = 2.0 × 10^−3^), C8 (*p* = 1.2 × 10^−2^), and C10:1 (*p* = 8.0 × 10^−3^) levels were significantly higher in the false-positive group, with minimal overlap between the groups. This indicates that elevated levels of these parameters of middle-chain acylcarnitines may help distinguish false positives from true positives. Notably, the highest values of C8, C10, and C8/C2 in the positive group (0.085, 0.126, and 0.0067, respectively) were exceeded by the lowest values of false-positive cases for each parameter (Figure 3). Consequently, 17 of the 19 false-positive cases could be distinguished from the true-positive cases using only three markers (Table 2 and Table 3).

#### 3.2.4. Comparison of Amino Acid Levels

No significant differences in amino acid levels were observed between the positive and negative control groups. However, when comparing the false-positive and negative groups, the Val (*p* = 2.0 × 10^−3^), Ala (*p* = 1.8 × 10^−2^), Leu (*p* = 2.6 × 10^−4^), Met (*p* = 2.8 × 10^−4^), and Orn (*p* = 5.1 × 10^−4^) levels were significantly lower in the false-positive group. The only significant difference between the positive and false-positive groups was observed for Val/Phe (*p* = 2.7 × 10^−2^), which was lower in the false-positive group. Eighteen of the 19 false-positive cases had lower Val/Phe values in comparison to the positive group (Table 4). These findings suggest that relatively low amino acid levels may serve as supportive indicators for false-positive results.

## 4. Discussion

Physiological weight loss in newborns is a common phenomenon in the first week after birth. All neonates typically lose approximately 5–10% of their birth weight within the first 3–5 days after delivery. This weight reduction primarily results from the diuresis of excess extracellular fluid and loss of meconium. Weight loss is also attributed to the redistribution of body water and the initial delay in establishing regular feeding, particularly in breastfed infants. Therefore, this weight loss is considered normal as long as the infants begin to regain weight and return to birth weight by approximately 10–14 days of age. Therefore, excessive or prolonged weight loss may indicate feeding difficulties, dehydration, or underlying metabolic concerns, which warrant further evaluation. In this study, the postnatal weight loss of newborns at the time of blood sampling was significantly greater in the false-positive group, which is consistent with previous reports [13]. The questionnaire survey also indicated a higher proportion of exclusively breastfed infants at the time of blood collection, suggesting that some breastfed infants may have had insufficient milk intake. Previous studies have also reported that breastfed infants experience significantly greater weight loss than formula-fed infants [3]. However, in the interviews of our study, some parents told us that their infants had received sufficient milk, as confirmed by weight measurements before and after feeding, yet they exhibited significant weight loss at the time of blood sampling. Under-recognized factors, such as the amount of glycogen stored at birth or maternal glucose tolerance during pregnancy, may also contribute to neonatal weight loss.

Regarding physical growth after discharge, the false-positive group exhibited catch-up growth at the 1-month follow-up visit. This indicates that there was no difference in growth (height and weight) between the false-positive and negative groups, despite significant weight loss at blood sampling, suggesting that transient weight loss did not affect subsequent physical growth.

Previous studies have reported that prolonged fasting (13–31 h) induces lipolysis, leading to increased long-chain acylcarnitines [15]. Additionally, the rate of weight loss in breastfed infants has been associated with elevated levels of long-chain acylcarnitines [16]. In the present study, the false-positive cases exhibited significant weight loss and elevation of a wide range of acylcarnitines. This suggests that starvation leads to increased β-oxidation of fatty acids, generating acetyl-CoA, which, in turn, results in elevated acylcarnitine levels. Previous reports have indicated that the ratio of C14:1 to medium-chain acylcarnitines significantly differed between false-positive and positive cases, suggesting its potential utility as a new marker [17]. In this study, we also demonstrated that medium-chain acylcarnitines were significantly elevated in the false-positive group in comparison to the positive group. Among medium-chain fatty acids, the C8, C10, and C8/C2 levels were substantially higher in the false-positive group than in both the positive and negative groups, highlighting their characteristic elevation in false-positive cases. Conversely, long-chain fatty acids, such as C12, C14, and C14:1, which are used as VLCADD screening markers, were markedly higher in the false-positive group than in the negative group. However, the C14:1/C2 and C14:1/C16 ratios tended to be higher in the positive group than in the false-positive group; therefore, their overlapping values meant that they could not exclusively distinguish between the two groups (Figure 3).

In the amino acid analysis, Val, Leu, and Orn tended to be lower in the false-positive group than in both the positive and negative groups. Previous studies have reported that small-for-gestational-age (SGA) infants exhibit reduced levels of amino acids such as Met, Phe, Tyr, and Leu on days 3–5 after birth [18]. Although the relationship between neonatal weight loss and amino acid levels remains unclear, it is plausible that insufficient amino acid supply after birth contributes to greater weight loss. In the present study, the false-positive group, that is, the group with lower body weight, exhibited a trend toward lower Val, Leu, and Orn levels relative to the negative group, suggesting a potential association with less intake. Notably, the Val/Phe ratio tended to be higher in the positive group than in the negative group. Thus, combining the findings on medium-chain fatty acids with Val/Phe values may serve as a supplementary index for distinguishing VLCADD-positive cases from false-positive cases. However, the present study included only three positive cases, and further investigation is required.

This study had several limitations. Perinatal maternal factors and fetal growth rates were not examined. Additionally, feeding conditions at the maternity hospital were assessed based on maternal interviews and maternal health record entries, suggesting that the actual feeding volume, feeding intervals, and maximum weight loss were not objectively measured. The timing of mass screening blood collection was not measured, and other factors, such as feeding status immediately before blood collection or the interval between feeding and blood sampling, may have influenced the results. Furthermore, it remains uncertain whether the one case with compound heterozygous variants represents a carrier or a mild VLCAD case. In addition, whether mild and atypical VLCAD cases might be included among the “19 false positives” cannot be excluded. Further validation is necessary to determine the appropriate cutoff values for differentiating positive cases.

## 5. Conclusions

This study confirmed that individuals with false-positive results for VLCADD in newborn screening exhibited significantly greater weight loss at the time of blood collection but did not experience adverse effects on subsequent growth. If an infant with significant weight loss at screening exhibits an elevation in VLCADD-specific acylcarnitines, this increase may, if transient, be caused by starvation, suggesting a potential for predicting false-positive results. The further analysis of acylcarnitine profiles, particularly elevated levels of C8, C10, and C8/C2, along with a low Val/Phe ratio, may be applicable for the identification of false-positive cases. Future research should include larger sample sizes to refine the cutoff values for these markers. This could ultimately enable tandem mass spectrometry for the rapid detection of VLDCAD in infants.

## Figures and Tables

**Figure 1 IJNS-11-00033-f001:**
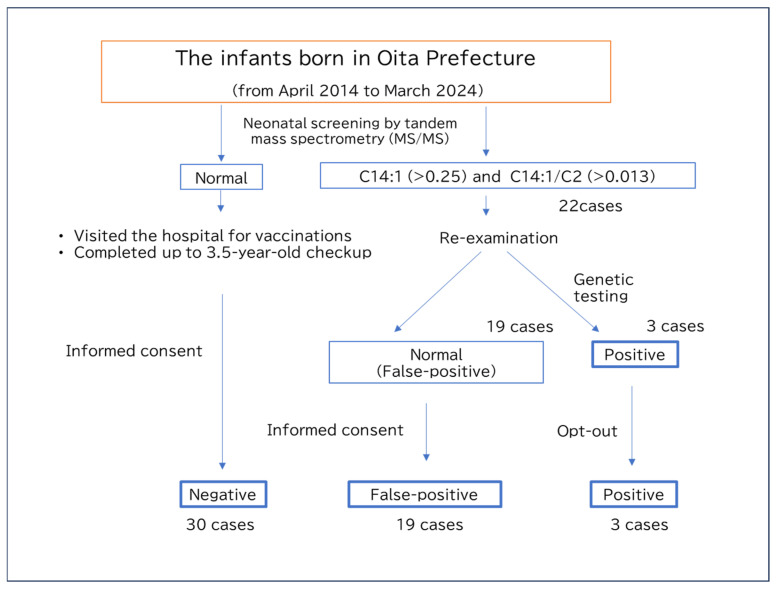
Flow chart of the present study.

**Figure 2 IJNS-11-00033-f002:**
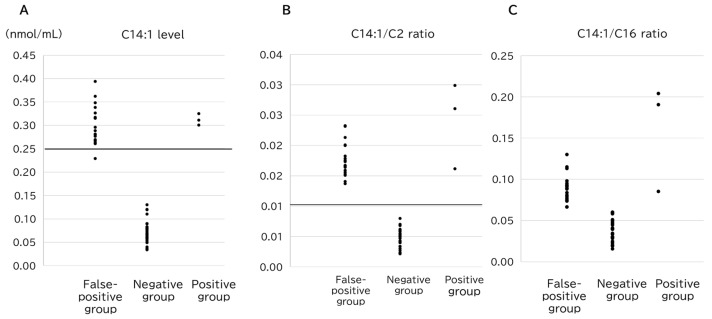
Comparison of the (**A**) C14:1 level, (**B**) C14:1/C2 ratio, and (**C**) C14:1/C16 ratio among the groups. The horizontal lines represent the cutoff value for VLCADD screening (C14:1, *y* = 0.25; C14:1/C2 ratio, *y* = 0.013).

**Figure 3 IJNS-11-00033-f003:**
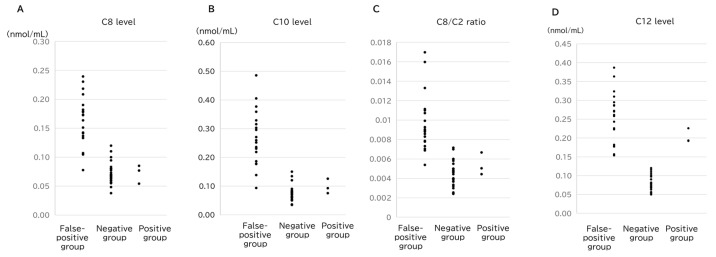
Comparison of the (**A**) C8 level, (**B**) C10 level, (**C**) C8/C2 ratio, and (**D**) C12 level among the groups.

**Table 1 IJNS-11-00033-t001:** Comparison of basic data and types of feeding among the groups.

		False-Positive (FP) Group	Negative (N) Group	Positive (P) Group	*p* Value
Number of participants		19	30	3	
Male:Female		12:7	18:12	2:1	
Gestational age	Weeks and days (median)	39w3d	39w6d	39w3d	
At birth	Weight (mean, g) (SD, g)	3061 (54.5)	3137 (65.5)	3052 (132.1)	0.703 *
	Height (mean, cm) (SD, cm)	48.8 (1.97)	49.5 (1.72)	N/A	0.377 *
At blood sampling	Postnatal (mean, days)	5	5	4.7	
	Weight (mean, g)	2811	3058	2946	0.030 *
	Weight reduction rate (%)	−9.2	−2.7	−3.9	
	Feeding method				
	exclusive breastfeeding (*n*)	6	4	N/A	
	mixed feeding (*n*)	5	26	N/A	
	formula feeding (*n*)	0	0	N/A	
	not assessed (*n*)	8	0	N/A	
1 month	Mean weight (mean, g), (SD, g)	3843 (500.1)	4144 (426.9)	N/A	0.061 **
	Weight-gain rate (g/day)	44.2	40.9	N/A	0.398 **
	Height (mean, cm) (SD, g)	52.7 (2.0)	53.1 (1.7)	N/A	0.450 **

* FP vs. P; ** FP vs. N; *n*, number; N/A, No analysis.

**Table 2 IJNS-11-00033-t002:** Comparison of acylcarnitine analysis between the groups.

Acylcarnitine	False-Positive Group (FP)	Negative Group (N)	Positive Group (P)	FP vs. P vs. N	FP vs. P	FP vs. N	P vs. N
Mean Level (95% C.I.) (nmol/mL)				*p* Value			
C0 ^(1)^ [<7.5]	13.0	17.3	15.5	2.3 × 10^−5^	1.9 × 10^−1^	3.8 × 10^−6^	3.5 × 10^−1^
	(10.5–14.9)	(16.2–20.8)	(13.7–17.9)				
C2	17.4	16.1	13.9	3.0 × 10^−1^	3.3 × 10^−1^	5.5 × 10^−1^	6.0 × 10^−1^
	(15.8–18.9)	(14.6–17.7)	(2.3–25.5)				
C3 ^(2)^ [>3.6]	1.2	1.3	1.4	4.8 × 10^−1^	3.8 × 10^−1^	2.9 × 10^−1^	7.0 × 10^−1^
	(0.92–1.5)	(1.2–1.6)	(1.3–1.5)				
C4	0.3	0.25	0.25	4.5 × 10^−2^	2.3 × 10^−1^	1.5 × 10^−2^	9.7 × 10^−1^
	(0.29–0.39)	(0.23–0.3)	(0.20–0.34)				
C5 ^(3)^ [>1.0]	0.1	0.13	0.12	7.0 × 10^−2^	2.6 × 10^−1^	2.4 × 10^−2^	9.7 × 10^−1^
	(0.083–0.12)	(0.11–0.14)	(0.12–0.14)				
C5:1 ^(4)^ [>0.025]	0.01	0.01	0.01	9.3 × 10^−1^	8.4 × 10^−1^	7.2 × 10^−1^	9.7 × 10^−1^
	(0.01–0.018)	(0.01–0.01)	(0.005–0.01)				
**C6**	0.08	0.053	0.04	8.1 × 10^−6^	**2.0 × 10^−3^**	**2.1 × 10^−5^**	4.5 × 10^−1^
	(0.069–0.091)	(0.048–0.059)	(0.0027–0.083)				
**C8** [>0.30] ^(5)^	0.17	0.07	0.08	2.1 × 10^−7^	**1.2 × 10^−2^**	**4.6 × 10^−8^**	8.9 × 10^−1^
	(0.14–0.19)	(0.06–0.08)	(0.065–0.085)				
**C8:1**	0.11	0.19	0.21	5.7 × 10^−6^	8.0 × 10^−3^	**6.8 × 10^−6^**	8.4 × 10^−1^
	(0.087–0.13)	(0.17–0.21)	(0.0060–0.40)				
**C10** [>0.4] ^(6)^	0.26	0.07	0.09	3.7 × 10^−8^	1.0 × 10^−1^	**5.0 × 10^−9^**	6.8 × 10^−1^
	(0.20–0.33)	(0.06–0.08)	(0.085–0.11)				
**C10:1**	0.1	0.063	0.063	1.4 × 10^−7^	**8.0 × 10^−3^**	**1.1 × 10^−7^**	1.0
	(0.090–0.11)	(0.056–0.069)	(0.012–0.11)				
C10:2	0.02	0.025	0.018	1.2 × 10^−1^	9.9 × 10^−1^	1.4 × 10^−1^	5.0 × 10^−1^
	(0.015–0.024)	(0.021–0.029)	(0.0059–0.043)				
**C12** [>0.4] ^(7)^	0.25	0.077	0.2	2.6 × 10^−17^	1.9 × 10^−1^	**5.1 × 10^−9^**	**4.7 × 10^−5^**
	(0.22–0.29)	(0.071–0.084)	(0.16–0.25)				
**C12:1**	0.2	0.02	0.06	1.2 × 10^−8^	2.2 × 10^−1^	**1.9 × 10^−9^**	8.5 × 10^−2^
	(0.16–0.24)	(0.02–0.03)	(0.06–0.1)				
**C14**	0.27	0.17	0.26	7.0 × 10^−8^	9.2 × 10^−1^	**7.3 × 10^−8^**	**1.5 × 10^−2^**
	(0.25–0.30)	(0.16–0.19)	(0.067–0.45)				
**C14:1^(8)^** **[>0.25]**	**0.3**	**0.073**	**0.31**	**1.0 × 10^−28^**	1.0	**5.1 × 10^−9^**	**5.1 × 10^−9^**
	**(0.064–0.082)**	**(0.064–0.082)**	**(0.28–0.34)**				
C14:2	0.04	0.02	0.09	2.3 × 10^−8^	7.5 × 10^−1^	**1.9 × 10^−9^**	**2.0 × 10^−3^**
	(0.04–0.05)	(0.01–0.02)	(0.06–0.095)				
C16 [>5.0] ^(9)^	3.6	2	1.6	3.5 × 10^−5^	4.8 × 10^−2^	**7.4 × 10^−6^**	8.5 × 10^−1^
	(2.9–3.9)	(1.6–2.5)	(1.6–2.6)				
C16:1	0.32	0.14	0.16	1.4 × 10^−6^	7.9 × 10^-2^	**2.1 × 10^−7^**	4.5 × 10^−1^
	(0.29–0.36)	(0.12–0.17)	(0.15–0.21)				
C18	0.86	0.63	0.89	2.6 × 10^−4^	9.5 × 10^−1^	4.0 × 10^−4^	5.7 × 10^−2^
	(0.77–0.94)	(0.57–0.70)	(0.096–1.68)				
C18:1	1.5	1.3	1	3.1 × 10^−2^	1.2 × 10^−1^	1.3 × 10^−2^	7.3 × 10^−1^
	(1.3–1.8)	(1.2–1.4)	(1.0–1.5)				
**C18:2**	0.15	0.29	0.14	**1.7 × 10^−5^**	1.0	**2.3 × 10^−5^**	**3.6 × 10^−2^**
	(0.13–0.17)	(0.25–0.33)	(0.080–0.21)				
**C5-DC**	0.13	0.084	0.12	4.7 × 10^−4^	8.8 × 10^−1^	**2.7 × 10^−5^**	1.3 × 10^−1^
	(0.11–0.14)	(0.07–0.94)	(0.010–0.25)				
C6-DC	0.095	0.08	0.09	3.2 × 10^−1^	6.7 × 10^−1^	2.1 × 10^−1^	2.9 × 10^−1^
	(0.07–0.11)	(0.06–0.1)	(0.085–0.14)				
**C4-OH**	0.25	0.11	0.17	3.1 × 10^−7^	3.5 × 10^−1^	**1.6 × 10^−7^**	3.8 × 10^−1^
	(0.20–0.29)	(0.09–0.12)	(0.21–0.56)				
C5-OH [>1.0] ^(10)^	0.13	0.08	0.13	7.5 × 10^−1^	6.3 × 10^−1^	6.8 × 10^−1^	4.8 × 10^−1^
	(0.11–0.14)	(0.07–0.1)	(0.095–0.15)				
C14-OH	0.025	0.01	0.01	5.3 × 10^−8^	3.7 × 10^−1^	**1.0 × 10^−8^**	5.6 × 10^−2^
	(0.02–0.03)	(0.01–0.01)	(0.01–0.02)				
C16-OH [>0.08] ^(11)^	0.03	0.01	0.02	4.6 × 10^−7^	5.2 × 10^−2^	**6.7 × 10^−8^**	2.4 × 10^−1^
	(0.023–0.03)	(0.01–0.02)	(0.015–0.025)				
C16:1-OH	0.037	0.031	0.033	2.1 × 10^−1^	8.0 × 10^−1^	1.8 × 10^−1^	9.7 × 10^−1^
	(0.032–0.041)	(0.027–0.036)	(0.0088–0.057)				
C18-OH	0.02	0.01	0.02	4.5 × 10^−7^	3.9 × 10^−1^	**5.6 × 10^−4^**	7.9 × 10^−2^
	(0.013–0.02)	(0–0.01)	(0.015–0.02)				
C18:1-OH [>0.05] ^(12)^	0.02	0.01	0.02	2.0 × 10^−3^	6.8 × 10^−1^	**8.7 × 10^−8^**	2.0 × 10^−1^
	(0.02–0.03)	(0.01–0.02)	(0.015–0.03)				

Marker cutoff levels for inborn errors of metabolism: (1) systemic primary carnitine deficiency, (2) methylmalonic acidemia/propionic acidemia, (3) isovaleric acidemia, (4) beta-Ketothiolase deficiency, (5) medium-chain acyl-CoA dehydrogenase deficiency, glutaric acidemia type 2, (6) glutaric acidemia type 2, (7) glutaric acidemia type 2, (8) very-long-chain acyl-CoA dehydrogenase deficiency, (9) carnitine acylcarnitine translocase deficiency, (10) methylcrotonylglycinuria, multiple carboxylase deficiency, 3-Hydroxy-3-methylglutaric acidemia, beta-Ketothiolase deficiency, (11) trifunctional protein deficiency, and (12) long-chain acyl-CoA dehydrogenase deficiency. Noteworthy analytes, as described in this manuscript, are indicated in **bold**.

**Table 3 IJNS-11-00033-t003:** Comparison of acylcarnitine ratios between the groups.

Acylcarnitine Ratio [Cutoff Value]	False-Positive Group	Negative Group	Positive Group	FP vs. P vs. N	FP vs. P	FP vs. N	P vs. N
	Mean Ratio			*p* Value			
C3/C2 [>0.22] ^(1)^	0.07	0.09	0.1	4.0 × 10^−3^	1.2 × 10^−2^	5.0 × 10^−3^	2.4 × 10^−1^
C4/C2	0.02	0.02	0.02	8.1 × 10^−1^	7.1 × 10^−1^	7.0 × 10^−1^	5.7 × 10^−1^
C4/C3	0.275	0.19	0.21	3.7 × 10^−2^	2.5 × 10^−1^	1.1 × 10^−2^	9.5 × 10^−1^
C4/C8	2.17	3.95	3.66	9.5 × 10^−6^	8.2 × 10^−2^	5.7 × 10^−6^	9.0 × 10^−1^
C5/C0	0.01	0.01	0.01	6.5 × 10^−2^	9.9 × 10^−1^	2.5 × 10^−2^	2.8 × 10^−1^
C5/C2	0.01	0.01	0.01	8.4 × 10^−4^	1.7 × 10^−2^	4.9 × 10^−4^	4.6 × 10^−1^
C5/C3	0.0873	0.0929	0.0907	7.9 × 10^−1^	9.8 × 10^−1^	7.7 × 10^−1^	9.9 × 10^−1^
C5OH/C8	1.39	3.15	2.73	2.9 × 10^−6^	9.6 × 10^−2^	1.6 × 10^−6^	7.7 × 10^−1^
C5DC/C8	0.84	1.20	1.60	4.7 × 10^−4^	3.0 × 10^−3^	4.0 × 10^−3^	1.5 × 10^−1^
C5DC/C16	0.0379	0.0421	0.0557	6.2 × 10^−2^	5.5 × 10^−2^	4.8 × 10^−1^	1.6 × 10^−1^
**C8/C2**	0.01	0.01	0.01	1.0 × 10^−7^	4.4 × 10^−2^	1.5 × 10^−8^	4.7 × 10^−1^
C8/C10 [>1.7] ^(2)^	0.643	1.060	0.745	5.1 × 10^−9^	6.5 × 10^−1^	8.8 × 10^−9^	1.9 × 10^−2^
**C14:1/C2** [>0.013] ^(3)^	0.0177	0.0047	**0.0241**	2.5 × 10^−23^	**6.5 × 10^−4^**	5.1 × 10^−9^	5.1 × 10^−9^
**C14:1/C16**	0.0893	0.0371	0.160	1.1 × 10^−15^	**1.7 × 10^−6^**	5.1 × 10^−9^	5.1 × 10^−9^
C16OH/C16	0.01	0.01	0.01	6.0 × 10^−3^	5.2 × 10^−1^	5.0 × 10^−3^	4.2 × 10^−2^
C0/(C16+C18) [>40] ^(4)^	2.76	7.87	6.71	1.2 × 10^−6^	1.2 × 10^−1^	1.7 × 10^−7^	3.3 × 10^−1^
(C16+C18:1)/C2 [>0.50] [>0.55] ^(5)^	0.294	0.217	0.251	2.1 × 10^−6^	2.5 × 10^−1^	1.1 × 10^−6^	4.2 × 10^−1^

Marker cutoff levels for inborn errors of metabolism: (1) propionic acidemia/ propionic acidemia, (2) medium-chain acyl-CoA dehydrogenase deficiency, (3) very-long-chain acyl-CoA dehydrogenase deficiency, (4) carnitine palmitoyl transferase I deficiency, and (5) carnitine palmitoyl transferase II. deficiency. Noteworthy analytes, as described in this manuscript, are indicated in bold.

**Table 4 IJNS-11-00033-t004:** Comparison of amino acid values.

Amino Acids	False-Positive Group	Negative Group	Positive Group	FP vs. P vs. N	FP vs. P	FP vs. N	P vs. N
	Mean Level (95% C.I.) (nmol/mL)			*p* Value			
**Val**	126.1	152.9	152.6	3.0 × 10^−3^	2.2 × 10^−1^	**2.0 × 10^−3^**	1.0
	(114.8–137.3)	(142.5–163.3)	(127.3–177.9)				
**Ala**	321.5	388.3	333.7	2.1 × 10^−2^	9.7 × 10^−1^	**1.8 × 10^−2^**	4.9 × 10^−1^
	(294.3–348.6)	(354.7–422.0)	(83.1–584.4)				
Arg	16.6	19	18.6	7.3 × 10^−1^	9.4 × 10^−1^	7.1 × 10^−1^	1.0
	(12.0–21.3)	(15.0–23.0)	(2.4–39.7)				
Cit	14.8	15.5	17.1	6.7 × 10^−1^	6.8 × 10^−1^	8.4 × 10^−1^	8.3 × 10^−1^
[>60] ^(1)^ [>85] ^(2)^	(12.0–17.5)	(14.1–17.0)	(8.4–25.9)				
Gly	477.7	435.75	423.7	2.0 × 10^−1^	2.6 × 10^−1^	9.7 × 10^−2^	7.3 × 10^−1^
	(439.8–543.3)	(388.1–496.2)	(412.6–448.7)				
**Leu**	128.6	177.7	168.7	4.0 × 10^−4^	2.2 × 10^−1^	**2.6 × 10^−4^**	9.2 × 10^−1^
	(112.0–145.3)	(162.5–192.9)	(49.0–288.3)				
**Met**	20.9	24.9	20.9	1.0 × 10^−3^	4.8 × 10^−1^	**2.8 × 10^−4^**	2.9 × 10^−1^
[>50] ^(3)^	(18.9–22.7)	(22.7–27.2)	(19.2–28.6)				
**Orn**	77.1	120.5	116	2.0 × 10^−3^	2.1 × 10^−1^	**5.1 × 10^−4^**	6.6 × 10^−1^
	(61.2–108.4)	(108.6–149.3)	(93.7–129.8)				
Phe	47.8	50.9	43.2	2.4 × 10^−1^	6.9 × 10^−1^	4.7 × 10^−1^	3.3 × 10^−1^
[>120] ^(4)^	(43.9–51.6)	(47.2–54.5)	(33.3–53.2)				
Pro	184.8	213.7	208.5	9.9 × 10^−2^	6.7 × 10^−1^	8.3 × 10^−2^	9.8 × 10^−1^
	(163.5–206.1)	(197.1–230.3)	(72.8–344.1)				
Tyr	122	98.6	78.3	6.8 × 10^−2^	3.4 × 10^−2^	1.2 × 10^−1^	1.6 × 10^−1^
	(91.8–153.9)	(84–117.8)	(71.7–83.8)				

Marker cutoff levels for inborn errors of metabolism: (1) citrullinemia, (2) argininosuccinic aciduria, (3) homocystinuria, and (4) phenylketonuria. Noteworthy analytes, as described in the results, are indicated in bold.

## Data Availability

The data analyzed during the current study are not publicly available to preserve individuals’ privacy. All data included in this study can be shared upon request to the corresponding author (k-ihara@oita-u.ac.jp).

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
