# Peer review of "Characteristic Findings of Infants with Transient Elevation of Acylcarnitines in Neonatal Screening and Neonatal Weight Loss"

_2409-515X, 2025, doi:10.3390/ijns11020033_

Round 1
Reviewer 1 Report
Comments and Suggestions for Authors
This is a very interesting study addressing the high false positive rates in newborn screening for VLCAD deficiency, also in relation to postnatal starvation. However, the manuscript would needs several improvements to be acceptable for publication.
Abstract - could be better structured, e.g. with subheadings.
Introduction - could be condensed, some parts are less important. Maybe better address the issue, how much is already known (any systematic reviews?) on this particular topic.
Methods and Results - probably too many figures and tables. Some tables (as cut-offs) could be moved to the supplementary materials. Questionnaire could be moved to the supplement. Some figures could be combined to form one bigger figure (e.g. with different C levels). Flowchart would better suit to Methods. Some parts of the Results should really be in the Methods section. Technical presentation of most of the Tables and Figures is qute poor. The data presentation is fragmented.
Comments on the Quality of English Language
Overal presentation of paper (technical, language, tables and figures) is quite poor.
Reviewer 2 Report
Comments and Suggestions for Authors
Comments to the Authors:
The authors present a study highlighting that elevated acylcarnitines—particularly C14 and C14:1—may result from postnatal starvation on the day of sampling, rather than an underlying inborn error of metabolism such as VLCADD. They analyzed data from neonates in Oita Prefecture over a 10-year period who were identified as false positives for VLCADD. The study proposed potential supplementary markers to help distinguish false positives from true VLCADD cases, including medium-chain acylcarnitines (C8, C10, and the C8/C2 ratio), as well as low levels of certain amino acids such as valine, leucine, and ornithine. The authors also address the parental anxiety caused by false positive results.
This study provides valuable insights into newborn screening and highlights the importance of standardizing the timing of blood spot collection—ideally before significant physiological weight loss occurs. It also serves as a reminder to carefully consider factors that may contribute to false positives. Here are few comments to consider.
Introduction:
Add reference for the definition of VLCADD and manifestations of VLCADD in neonatal period (line 48-51).
Materials and Methods
2.1 Neonatal Screening System in Oita
Regarding the statement: “Genetic testing was conducted as needed”, part of the protocol of VLCADD NBS is to obtain genetic testing for all initial postive screens per ACMG guidelines.
https://www.acmg.net/ACMG/Medical-Genetics-Practice-Resources/ACT_Sheets_and_Algorithms.aspx
Results
3.1 Analysis of Subjects
Line 166: “There were no significant differences in growth between the false posisitve and false negative groups at .…”. Do you mean negative group?
3.2 Questionnaire Survey
Line 170: “Both the false-positive and false-negative groups …”. Do you mean negative group?
Line 176: “false-negative group”. Do you mean negative group? Please confirm.
Line 184: “false-negative infants”. Do you mean negative group?
Were you able to genotype the false positive group to see whether the babies were heterozygous carries for a pathogenic variant in ACADVL gene?
3.2.3. Comparison Between False-positive and Positive Groups
representative VLCAD marker, C14:1, was markedly elevated in both the false-positive and false-positive groups in comparison to the normal subjects, ….”. Please correct the duplicate, I believe you mean false-positive and positive.
3.2.4. Comparison of Amino Acid Levels (Table 7)
Line 268: “However, when comparing the false-positive and false-negative groups, ….”. False negative or negative?
Discussion:
Line 303: “growth (height and weight) between the false-positive and false-negative groups, despite..” Please correct.
Line 332: “significantly differed between false-positive and false-positive cases, suggesting its pote..”. Please correct.
While significant weight loss at the time of newborn screening (NBS) collection could potentially lead to false positive elevations in C14:1 and C14:1/C2 ratios, NBS blood spots are typically collected within 48–72 hours after birth—before the peak period of physiological weight loss. Including the exact age of neonates at the time of blood spot collection for both false positive and true positive cases would provide valuable context.
Incorporating molecular genetic testing as a second-tier test within the NBS protocol for VLCADD-positive cases is crucial. A normalized acylcarnitine profile alone is insufficient to exclude VLCADD, as acylcarnitine levels can be influenced by the infant’s metabolic state. Several challenges complicate VLCADD screening and diagnosis: acylcarnitine abnormalities may normalize under physiologically stable conditions, such as post-feeding or after glucose administration, increasing the risk of misclassifying milder phenotypes.
Tables:
Regarding the tables. There are number at the end of the acycarnitine and aminoacids that are not defined at the Table legends. Please explain them.
For example:
Table 5. C8/C102)
Table 7. Val1)
Table 2. Comparison of the physical growth between false negative and negative groups up to age of 3 years.
Do you mean false positive?
Table 3. Comparison of the questionnaire between false negative and negative groups.
Do you mean false positive?
Table 4 and Table 5. It will be clear if you can add the cutoff valuaes for each acylcarnitne and ratios in the table (below each acylcarnitine.
Table 7. Define the abbreviations.
Figure 2. y=0.025 or 0.25?
Reviewer 3 Report
Comments and Suggestions for Authors
Thank you very much for the opportunity to review this manuscript. The reduction of false positive results for VLCAD deficiency is an ongoing issue for NBS laboratories and additional strategies are always needed. This work describes in detail the association between increased weight loss between birth and DBS collection with false positive results for VLCAD. I think this observation and data are valuable and may offer additional paths for clinicians to sufficiently resolve VLCAD screens.
I have a few general comments and then some minor corrections as well:
- Was sequencing done for any of the FP results to confirm if they were carriers or not? This is also a confounding factor for VLCAD NBS results and should be identified if present.
- What method was used for acylcarnitine analysis? Derivatized vs underivatized? This is important for the implications of the C5DC acylcarnitine described.
- Line 176: "False negative group", I believe this should just be the "negative" group
- Table 2 may be more understandable as a chart
- For Table 4 and 5 - a focus on the key analytes or at minimum the ones that show a significant difference between FP results may make the message more impactful. Alternatively, highlighting those that reach statistical significance may be helpful. The key message of the paper (weight loss / starvation immediately post birth can result in FP screening results for VLCAD deficiency) is getting swamped in the large amount of data that does not show any difference between the groups.
- Linked back to the comment about which method is used for acylcarnitine analysis - if underivatized, consider the possibility that the elevation of C5DC is actually an elevation of C10OH.
- When describing the impact of elevations of medium chain acylcarnitines, was any of this correlated to formula intake and was it possible the formula included medium chain triglycerides?
- Figures 2-5 show minimally helpful information, and may be better presented in a streamlined manner. Alternatively, if they stay the same - acylcarnitine ratios are unitless, rather than nmol/mL.
- Table 6 is not necessary, although including the VLCAD algorithm is helpful. Possibly in the Methods when describing the identification of the screening population.
- Line 268 again refers to "false negative", when I think this should be "negative" or "true negative"
- Table 7 shows a lot of data that is not significant (per the description in the text). Again, the main message of the paper is getting swamped with a lot of data that is non-contributory to the overall conclusion. The conclusion and supporting data for this paper is important, but ultimately hard to find in the current presentation of the paper.
Overall, I think consideration of weight loss in the NBS setting is an interesting and exciting low impact way to improve screening. I think tightening up this manuscript to clearly focus on those findings has the potential to help in a problematic area.
Round 2
Reviewer 1 Report
Comments and Suggestions for Authors
The manuscript is now much improved. I think it is suitable for publication in present form.
Comments on the Quality of English Language
I suggest further language and technical improvements of the manuscript.
Reviewer 3 Report
Comments and Suggestions for Authors
Thank you for the thorough and detailed response to the comments raised in the initial review. All of my substantial concerns about the content of the article have been addressed. From a minor formatting perspective, I would note there are some font differences throughout, but the content has all been adequately addressed. I think the tightened focus of the article makes it much better at getting the main point across.